# Association of Antiphospholipid Antibodies with Clinical Manifestations in Children with Systemic Lupus Erythematosus

**DOI:** 10.3390/jcm12041424

**Published:** 2023-02-10

**Authors:** Gordana Petrovic, Srdjan Pasic, Ivan Soldatovic

**Affiliations:** 1Mother and Child Health Care Institute, 11000 Belgrade, Serbia; 2School of Medicine, University of Belgrade, 11000 Belgrade, Serbia

**Keywords:** pediatric rheumatology, antiphospholipid antibodies, childhood SLE, SLEDAI

## Abstract

Background: The aim of the study is to evaluate the effect of the presence of antiphospholipid antibodies on the clinical and laboratory manifestations, disease activity and outcomes of the disease in patients with childhood-onset systemic lupus erythematosus (cSLE). Methods: We conducted a 10-year cross-sectional study with a retrospective analysis of clinical and laboratory parameters and outcome of the disease (kidney, nervous system involvement, thrombosis). For the purpose of the study, patients were divided into cohort groups based on the presence of antiphospholipid antibodies (aPLA), named the aPLA positive group, or their absence, named the aPLA negative group. Values of aPLA were defined in reference laboratories. The disease activity was measured by the Systemic Lupus Erythematosus Disease Activity Index 2000 (SLEDAI-2K) score, whereas tissue damage degree was measured by Systemic Lupus International Collaborating Clinics/American College of Rheumatology-Damage Index (SLICC/ACR DI; SDI; DI). Results: Research in our center showed that patients with cSLE often had hematological, cutaneous, and non-thrombotic neurological manifestations. Antiphospholipid antibodies may be present transiently or permanently. A significant change in the titer value was observed in the IgG isotype of aCLA. The presence of higher values of IgM β2GP1 at the beginning indicates that higher disease activity can be expected. Higher disease activity correlates with greater tissue damage. Additionally, it has been shown that aPLA positive patients have two and a half times higher risk of tissue damage than aPLA negative ones. Conclusion: Our study shows that the presence of antiphospholipid antibodies in patients with childhood onset systemic lupus erythematosus may indicate a higher risk of tissue damage, but since it is a rare disease in childhood, prospective and multicenter studies are necessary to assess the importance of the presence of these antibodies.

## 1. Introduction

Systemic Lupus Erythematosus (SLE) is a multisystem autoimmune disease with onset in childhood (cSLE) in 10–15% of cases [1,2,3,4] and is associated with significant morbidity due to the disease itself causing multiple end-organ damage and as well due to drug side effects [4,5,6,7,8]. Antiphospholipid syndrome (APS) is another systemic autoimmune disease characterized by vascular thrombosis in the presence of aPLA on two or more occasions at least 12 weeks apart [9]. Antiphospholipid antibodies include lupus anticoagulants (LA), anticardiolipin antibodies (aCLA), and anti-β2glycoprotein I antibodies (β2GPI). Primary APS is rare in childhood [10,11], but the frequency of aPLA presence is high in cSLE, reported in ~40% and associated with older age at onset and higher frequency of venous thrombosis and hematologic and skin involvement [12,13,14,15,16,17,18,19,20,21,22,23]. Related clinical manifestations (e.g., neuropsychiatric, renal, hematological) have been reported in aPLA positive children, but the influence of aPLA presence on the disease course remains unclear. We conducted a retrospective study that included a 10-year follow-up of 40 patients with cSLE where we assessed the impact of aPLA on clinical manifestations, disease activity, and the occurrence of tissue and organ damage.

## 2. Materials and Methods

This research was conducted in a tertiary pediatric center, approved by the ethics commission of the institution, with informed consent of the patients. It included children diagnosed in the period from January 2011 to December 2020. The course of the disease was followed up over the period of 10 years through clinical and laboratory parameters. The research included patients who met the criteria of the American College of Rheumatology (ACR, 1997) and Systemic Lupus International Collaborating Clinics (SLICC, 2012) for the SLE diagnosis [1,2]. The eliminatory criterion was age above 17 at the time of diagnosis due to the inability of further follow-up. We divided the subjects into a cohort group that at the moment of diagnosis of systemic lupus erythematosus had aPLA, the aPLA positive group, and a group without antibodies, the aPLA negative group. The research was designed as a cross-sectional study with a retrospective analysis of the baseline clinical and laboratory parameters of the disease and outcomes (hematological and skin manifestations, lupus nephritis, central nervous system disorder, thrombosis). For the purpose of the study, complete blood count, coagulation factors, and antiphospholipid antibodies were tested. Measurements were performed at the time of diagnosis of the systemic lupus erythematosus, before starting treatment, and after one and three years of the disease follow-up. The disease activity was measured by the Systemic Lupus Erythematosus Disease Activity Index 2000 (SLEDAI-2K) score, whereas tissue damage degree was measured by the Systemic Lupus International Collaborating Clinics/American College of Rheumatology-Damage Index (SLICC/ACR DI; SDI; DI). Positive aPLA values were checked after 12 weeks to confirm positivity.

The lupus anticoagulants (LA) in the samples of each enrolled patient were tested and positive values were ≥1.2. For the detection of LA antibodies, the dilute Russel viper venom test (DRVVT) was applied. Anticardiolipin and β2GP1 antibodies were tested by Enzyme Linked Immuno Sorbent Assay (ELISA) technique (Euroimmun, Mediziniche Labordiagnostika AG) and the results were expressed in international units (PLU/mL). They were defined as positive if aCLA (IgM/IgG) was >12 PLU/mL. Values less than 20 PLU/mL were weakly positive, from 20 to 40 PLU/mL moderately positive, and above 40 PLU/mL strongly positive. Anti-β2glycoprotein I antibodies (β2GPI) positive values were those above 10 U/mL. The control group consisted of patients with cSLE without antibodies, who were aPLA negative.

Relative number measures (structure indicators) were used as descriptive statistical methods, and for testing statistical hypotheses: *t*-test for two dependent and independent samples, Wilcoxon test, Chi-square test, and Fisher’s exact test. We used odds ratio (OR) as a measure of association and the comparison between groups of patients was made by univariate analysis (logistic regression analysis), whilst the correlation analysis was used for examining the correlation of the relevant variables. To determine the probability of occurrence of complications during the 10-year follow-up period, modified Kaplan–Mayer curves were used. The Long Rank test was used to examine the difference in the likely occurrence of observed complications between the aPLA positive and aPLA negative patients. Spearman’s correlation coefficient was used to examine the association between SLEDAI score values and observed antibodies. Statistical data analysis was performed using statistical software Easy R v.1.40. Statistical hypotheses were tested on the level of statistical significance (alpha level) of 0.05.

## 3. Results

The final group consisted of 40 patients, 31 female and 9 male, 17 aPLA positive and 23 aPLA negative. The demographic characteristics of the respondents are shown in Table 1. All the observed demographic characteristics were comparable among the formed groups of respondents. Manifestations during the follow-up are summarized in Table 2. The most frequent associated manifestations at the beginning were hematologic disorders that were present in 11 (64.7%) in the aPLA positive, and 15 (65.2%) patients in the aPLA negative group, followed by skin disorders in 8 (47.1%) aPLA positive and 7 (30.4%) aPLA negative patients, and nonthrombotic neurologic disorders in 2 (11.8%) aPLA positive patients. After three years there was a decrease in the number of patients with hematological and an increase in the number of cutaneous and neurological manifestations in aPLA positive group.

Deep vein thrombosis of the leg, in the second year of the disease, was developed by one patient. This patient had a permanent presence of LA, and aCL and anti β2GP1 antibodies were detected at the time of thrombosis. Regarding talk about kidney and CNS involvement, no statistically significant differences were observed between the observed groups. Kaplan–Mayer curves were used to assess the probability of disease occurrence in the observed groups (Figure 1 and Figure 2).

The obtained results show us a slightly higher probability of lupus nephritis in the aPLA negative group of patients, which was also the highest in the first four years of follow-up, but with the possible occurrence of this type of complication in the later period.

Of the six children with CNS manifestations in our study, two had vasculitis, three had psychosis, and one patient had chorea. All children were aPLA positive; all patients had aCLA, the patient with chorea had LA, and the children with vasculitis had both LA and β2GP1. At the time of diagnosis, one of the aPLAs had four patients (23.52%), two antibodies had 10 (58.8%), and two children (11.76%) had all antiphospholipid antibodies.

After three years, four patients (23.52%) had one antibody, six (35.29%) had two, and three (17.64%) had all three antibodies. In our study, at the beginning of the disease, elevated values of LA antibodies were present in 6/17 patients (35.3%), and after three years in 10/17 patients (58.82%). At the time of diagnosis, aCLA was not performed in 1 patient, elevated class IgM was present in 11/16 patients (68.75%), and IgG in 10/16 patients (62.5%), and after follow-up with 15 patients (two follow-ups were not performed), 9 patients (60%) had an elevated IgM and IgG class after three years. At the beginning, β2GP1 was not performed in 2 patients, and 5/15 (33.33%) and 3/15 patients (20%) had elevated IgM class. After three years, of 12 patients, elevated IgM antibodies were registered in 5 (41.66%), and IgG class in 2 (16.66%) patients. The aPLA values of our patients are given in Appendix A.

No significant decrease in aPLA values was observed in the third year of follow-up except IgG class aCLA (*p* = 0.0437) (Table 3). By analyzing the association of antiphospholipid antibody values with the SLEDAI score, a statistically significant association was observed only at the onset of the disease between the SLEDAI score and the β2GP1 IgM value (Table 4). These values of the correlation coefficient indicate that higher values of SLEDAI score are expected in subjects with higher values of beta2GP1 IgM.

We compared the prognostic impact of aPLA considering the SLEDAI and DI. In the aPLA negative group, the SLEDAI score was reduced in 19/23 patients (82.6%), and in the aPLA positive group in 9/17 patients (52.9%). A DI above one in the aPLA negative group initially had 1/23 patients, and after follow-up, it had 5/23 (21.7%), while in the aPLA positive group initially 1/17 patients, and after three years it was 6/17 (35.3%). In the aPLA positive group mean DI was 3.6, whereas it was 1.4 in aPLA negative patients. In aPLA positive patients the risk of damage was two and half times as high as in aPLA negative patients. The more frequent systems affected were renal (60%) and neuropsychiatric (5%).

At the time of disease diagnosis, one patient with tissue damage measured by DI was recorded in each group. The aPLA positive group had a higher number of patients with damages in the first and third year (seven and six patients respectively), whilst the aPLA negative group had four patients with damages.

By comparing SLEDAI-2K and DI, it was concluded that there was a statistically significant medium correlation between these two scores in both groups at the third year of follow-up. The higher the disease activity, the more severe tissue damage consequently is (Table 5).

## 4. Discussion

Studies show that 38% to 75% of patient with cSLE have at least one form of aPLA; 19–87% have aCLA, about 40% have β2GP1, and from 10% to 62% have LA [12,13,14,15,16,17,18,19,20,21,22,23]. Numerous papers report the presence of various clinical manifestations in aPLA positive children with SLE. Our data showed that patients with antibodies had more frequent hematological, cutaneous, and non-thrombotic neurological manifestations at the beginning and during the follow-up period, which coincides with the results of other researchers [10,15,24]. Some studies have examined the association of aPLA with neuropsychiatric manifestations in pediatric SLE. In a Canadian study, these manifestations were observed in 26% of patients during the three-year follow-up, and the prevalence of anti-β2GP1 antibodies was found to be significantly higher in the group of patients with neuropsychiatric disease as compared to those without [21]. According to the literature, chorea has been described as an isolated clinical manifestation in aPLA positive patients with SLE [21,24,25,26,27]. No clear association was found between aPLA and migraines [28].

Dermatological manifestations of aPLA have not been studied in detail so far. One study found that 21% of children with the Raynaud phenomenon have aPLA [29]. According to the results of Campos et al., no difference was observed in the frequency of mucocutaneous manifestations between aPLA positive and aPLA negative patients with lupus [12]. Raynaud phenomenon, migraine headaches and chorea, and hematologic disorders are seen more frequently in children compared to adults with aPLA [9].

A meta-analysis of studies investigating the frequency and clinical significance of aPLA in cSLE showed a prevalence of 44% for aCLA, 40% for anti β2GP1, and 22% for LA. Regarding the isotypes antibody, the aCLA was more frequent than LA and anti β2GP1 [15]. Similar frequencies were observed in our study.

Some pediatric studies suggested a significant association of positive aPLA with SLE disease activity and damage [4,12,21,25]. The results of our research suggest that in patients with cSLE who have higher IgM isotype β2GP1 values at the beginning of the disease, higher values of SLEDAI score can be expected. Additionally, higher disease activity indicates consequent greater damage during follow-up. These conclusions are consistent with the results obtained by some other researchers. Brunner et al. investigated the relationship between disease activity and disease damage in 66 prospectively followed children with SLE and found that cumulative disease activity over time was the single best predictor of the development of disease damage [4]. Campos et al. did not find a linear correlation between IgG or IgM aCLA antibodies and the activity of SLE in their research, when considering the mean aCLA and mean values of SLEDAI [12]. However, some pediatric studies suggested that aCLA values may change with changes in SLE disease activity. A follow-up study of 137 patients with cSLE showed positive correlation of aCLA titers with SLEDAI [21]. Finally, our study showed that patients with cSLE, aPLA positive, had a two and half times higher risk of tissue damage compared to aPLA negative patients, which is similar to the results obtained by French researchers proving that the presence of aPLA tripled the risk of damage [25].

## 5. Conclusions

Research in our center showed that patients with cSLE often had hematological, cutaneous, and non-thrombotic neurological manifestations. Antiphospholipid antibodies may be present transiently or permanently in these patients. A significant change in the titer value was observed in the IgG isotype of aCLA. The presence of higher values of the IgM isotype β2GP1 at the beginning of the disease indicates that higher disease activity can be expected, which correlates with greater tissue damage. Additionally, it has been shown that aPLA positive patients have two and a half times higher risk of tissue damage than aPLA negative ones. Considering that it is a rare disease in childhood, prospective and multicenter studies are needed to determine the significance and impact of antiphospholipid antibodies on clinical manifestations, course, and possible occurrence of complications and tissue damage in children with SLE.

## Figures and Tables

**Figure 1 jcm-12-01424-f001:**
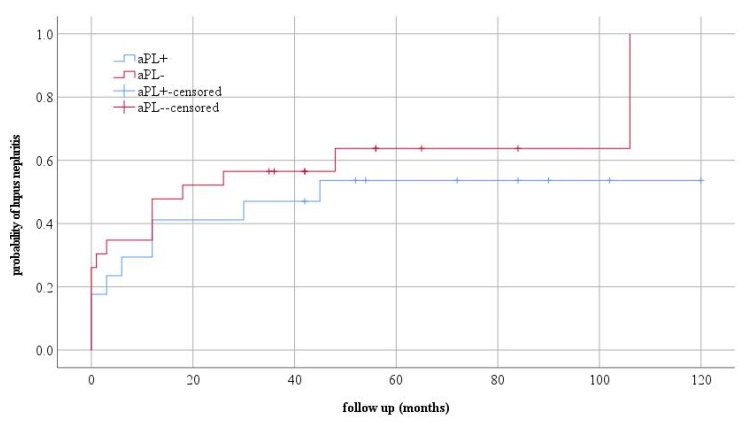
The probability of the occurrence of lupus nephritis and the clinical form of the disease.

**Figure 2 jcm-12-01424-f002:**
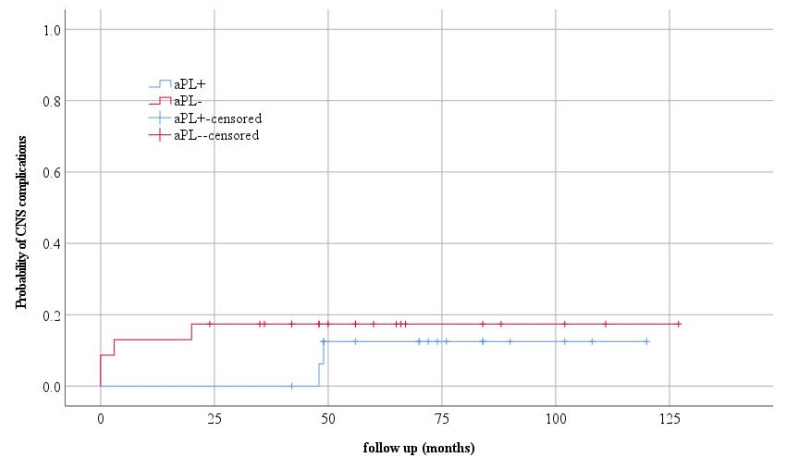
The probability of the occurrence of CNS complications and the clinical form of the disease.

**Table 1 jcm-12-01424-t001:** Demographic characteristics of the respondents.

Observed Variables	All Patients	aPLA Positive	aPLa Negative	Exp B (OR) 95% CI
Gender, girls n (%)	31/40 (77.5%)	11/17 (64.7%)	20/23 (87%)	0.275 (0.057–0.321)
Age at onset of the disease, average (SD)	12.35 (2.48)	12.94 (1.89)	11.91 (2.80)	0.835 (0.634–1.099)
Age at the time of diagnosis, average (SD)	12.40 (2.70)	13.00 (1.90)	11.96 (3.13)	0.855 (0.663–1.104)
Observation time, med (min-max) months	68.50 (24–127)	75.41 (42–120)	66.83 (24–127)	0.986 (0.962–1.012)

Comparison of groups with different clinical picture; univariate logistic regression.

**Table 2 jcm-12-01424-t002:** Presentation of clinical manifestations in the group of aPLA positive and aPLA negative patients.

Clinical Manifestation	aPLA Positive			aPLA Negative No of Patients		
	Beginning	1 Year	3 Years	Beginning	1 Year	3 Years
**Hematological disorders**						
Thrombocytopenia	8	4	4	7	5	5
Leucopenia	5	1	2	9	4	2
Autoimmune hemolytic anemia	7	3	1	12	9	4
**Skin disorders**						
Livedo reticularis	3	5	9	3	6	11
Raynaud phenomen	5	8	11	4	7	12
**Neurologic disorders**						
Migraine headache	1	4	2	0	2	1
Epilepsy	0	1	1	0	0	0
Mood disorders	1	1	3	0	1	1

**Table 3 jcm-12-01424-t003:** Change in aPLA’s value during the follow-ups.

Group	Type Antibody	Period	Number (N)	AS ± SD	Median	25 Percent	75 Percent	*p*-Value
**With** **aPLA**	LA	beginning	17	1.27 ± 1.30	1.10	0.81	1.30	0.538 ^**a**^
		1 year	17	1.42 ± 0.56	1.10	0.92	1.80	
		3 year	17	1.52 ± 0.68	1.38	0.98	1.98	
	β2GPI IgM	beginning	15	17.67 ± 11.45	16.00	6.30	31.00	0.441 ^**b**^
		1 year	13	31.00 ± 51.92	14.60	11.20	26.10	
		3 year	10	12.62 ± 7.45	13.00	6.20	26.70	
	β2GPI IgG	beginning	15	10.77 ± 7.70	7.20	4.30	19.20	1.000 ^**a**^
		1 year	13	9.83 ± 7.05	6.90	5.10	12.30	
		3 year	10	10.73 ± 9.14	6.25	5.00	14.17	
	ACLA IgM	beginning	16	24.94 ± 28.14	19.20	8.30	25.15	0.151 ^**a**^
		1 year	15	25.80 ± 49.45	12.20	7.65	16.10	
		3 year	15	14.40 ± 12.60	12.60	7.30	17.30	
	ACLA IgG	beginning	16	25.15 ± 27.72	14.45	8.15	31.20	0.0437 ^**a**^
		1 year	15	11.40 ± 7.50	10.80	5.55	16.60	
		3 year	15	12.68 ± 8.39	14.20	4.50	18.75	

^**a**^ Wilcoxon test for two dependent samples, ^**b**^
*t*-test for two dependent samples.

**Table 4 jcm-12-01424-t004:** Correlation of aPLA with SLEDAI score during three-year follow-up.

Period of Follow-Up	Antibody	SLEDAI Score
**Beginning**	LA	ρ = −0.145; *p* = 0.386
	aCLA IgM	ρ = 0.054; *p* = 0.747
	aCLA IgG	ρ = 0.170; *p* = 0.307
	Beta2 GPI IgM	ρ = 0.352; *p* = 0.035 *
	Beta2 GPI IgG	ρ = 0.107; *p* = 0.541
**1 year**	LA	ρ = −0.263; *p* = 0.110
	aCLA IgM	ρ = 0.096; *p* = 0.595
	aCLA IgG	ρ = 0.276; *p* = 0.121
	Beta2 GPI IgM	ρ = 0.341; *p* = 0.070
	Beta2 GPI IgG	ρ = −0.133; *p* = 0.492
**3 year**	LA	ρ = 0.122; *p* = 0.467
	aCLA IgM	ρ = −0.160; *p* = 0.415
	aCLA IgG	ρ = −0.151; *p* = 0.443
	Beta2 GPI IgM	ρ = 0.308; *p* = 0.134
	Beta2 GPI IgG	ρ = 0.243; *p* = 0.263

* Statistically significant association; Spearman’s correlation coefficient.

**Table 5 jcm-12-01424-t005:** Association of SLEDAI-2K with DI during three-year follow-up.

Period of Follow-Up	Patient Groups	*p*-Value	Spearman’s Correlation Coefficient
Beginning	aPLA negative	0.410	−0.116
	aPLA positive	0.208	0.836
1 year	aPLA negative	0.508	−0.145
	aPLA positive	0.102	0.410
3 year	aPLA negative	0.001	0.635
	aPLA positive	0.006	0.632

*t*-Test for two independent samples; Spearman’s correlation coefficient.

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
