# Peer review of "Association of Antiphospholipid Antibodies with Clinical Manifestations in Children with Systemic Lupus Erythematosus"

_jcm, 2023, doi:10.3390/jcm12041424_

Round 1

Reviewer 1 Report

Petrovic et al evaluated the effect of the presence of antiphospholipid antibodies on the clinical and laboratory findings, disease activity and outcome in children with systemic lupus erythematosus (SLE) in a ten-year cross-sectional study with a retrospective analysis. In this study, children with SLE often had hematological, cutaneous and non-thrombotic neurological manifestations. Higher disease activity correlates with greater tissue damage. aPL positive patients have two and a half times higher risk of tissue damage than aPL negative ones. This report will be valuable because SLE in children is rare and there are few coherent reports. However, due to the small number of cases, the interpretation of the results should be done with great caution. I have a few questions regarding these points.

major concerns)

1) In Table IV, b2GPI IgM and IgG were only analyzed. How about the correlation between LAC (and/or aCLA) and SLEDAI score? Please provide this information for the reference of other readers.

2) I may have missed it, but do the correlations between aPLA and SLEDAI in Table IV and SLEDAI-2K and DI in Table V mean that they are both correlations of scores at each time point? Or do they mean that the aPLA value at the initial visit correlates with the SLEDAI at each time point? Do you mean that the SLEDAI-2K at the initial visit correlates with the DI? It would be easier to understand if you could clarify this point in the title of each table.

3) In line 22, you state, "it has been shown that aPLA positive patients have two and a half times higher risk of tissue damage than aPL negative ones." In Graphs 1 and 2, LN and CNS lupus tend to be more common in aPL- patients. Both data could be considered contradictory. That is, in the former, aPL+ can be interpreted as milder and in the latter, aPL+ as more severe. How do you explain this divergence in the data? One possibility that I have considered is that the aPL+ patients may have been treated more aggressively from the beginning because they had various organ lesions at the time of their first visit. This may have resulted in less tissue damage during follow-up. The current paper does not describe the details of the treatment, but I would appreciate it if you could take this into consideration and provide information that would be useful for the flow of the paper and for clinical practice.

minor concerns)

1) In line 35, what does "c-SLE" stand for, do you mean clinical SLE? If so, please add the abbreviation to the first occurrence for clarity. Also, since both cSLE and c-SLE appear, please unify the notation.

2) In line 81, "and15", please put a space between "and" and "15".

3) In table 1, "SEL", do you mean SLE? Please correct accordingly.

4) In line 97, "Graph 1, 2" should be written in accordance with the submission rules, such as "Figure" instead of "Graph".

5) In Table I, II, 3, IV, V, numbers are not unified.

6) In Table IV, "SLEDAI scor" is considered to mean "score".

Author Response

Dear Professor,
Thank you very much for all the suggestions. I tried to answer the questions and ambiguities. Also, since the correction of the English language was required, I requested the help of the official service of the journal platform (English Editing). I hope that the corrected text is more understandable. I am sending the following documents in the attachment:
1) response to your comments, major and minor
2) corrected version of the text
3) proposal of modified tables and figures
4) English Editing Certificate

Thank you very much for all the information and help in reviewing my text.

Kind regards

Gordana Petrovic

Response to Reviewer 1 Comments

Major concerns

Point 1 In Table IV, b2GPI IgM and IgG were only analyzed. How about the correlation between LAC (and/or aCLA) and SLEDAI score? Please provide this information for the reference of other readers.

Response 1

Yes, the correlation of other antiphospholipid antibodies with the SLEDAI score was also done. I presented the data in a table. If you think there is a more correct or clearer way to display these results, I would be very grateful if you could suggest it.

According to your suggestion, I tried to give a different title to Table IV. I don't know if it's clearer now. Thank you very much for your help.

And, I would like to apologize for the fact that I will send these tables in another, additional word document. Namely, during the previous shipment, some parts were lost (as in Table II). I'm sorry if I'm causing a problem, I just want to convey the complete information correctly. Thank you in advance for your understanding.

Table IV Correlation of aPLA values with SLEDAI score during three-year follow-up

period of follow-up

Antibody

SLEDAI score

At the disease onset

LA

ρ=-0,145; p=0,386

aCLA IgM

ρ=0,054; p=0,747

aCLA IgG

ρ=0,170; p=0,307

Beta2 GPI IgM

ρ=0,352; p=0,035*

Beta2 GPI IgG

ρ=0,107; p=0,541

After the first year

LA

ρ=-0,263; p=0,110

aCLA IgM

ρ=0,096; p=0,595

aCLA IgG

ρ=0,276; p=0,121

Beta2 GPI IgM

ρ=0,341; p=0,070

Beta2 GPI IgG

ρ=-0,133; p=0,492

After the third year

LA

ρ=0,122; p=0,467

aCLA IgM

ρ=-0,160; p=0,415

aCLA IgG

ρ=-0,151; p=0,443

Beta2 GPI IgM

ρ=0,308; p=0,134

Beta2 GPI IgG

ρ=0,243; p=0,263

*statistically significant association; Spearman's correlation coefficient

Point 2 I may have missed it, but do the correlations between aPLA and SLEDAI in Table IV and SLEDAI-2K and DI in Table V mean that they are both correlations of scores at each time point? Or do they mean that the aPLA value at the initial visit correlates with the SLEDAI at each time point? Do you mean that the SLEDAI-2K at the initial visit correlates with the DI? It would be easier to understand if you could clarify this point in the title of each table.

Response 2

Dear Professor,

Thank you for this suggestion. I tried to clarify, I hope that I will succeed in this, and that it will be clearer to my colleagues.

The research was conducted on a group of patients whose systemic lupus erythematosus started in childhood (chidhood onset systemic lupus erythematosus, cSLE). At the beginning of the research, two groups of respondents were formed. One included patients with cSLE who had positive aPLA (aPLA positive group). The second group consisted of patients who did not have aPLA (aPLA negative group).

In aPLA positive patients, a correlation of aPLA with the SLEDAI score of these patients was made at the beginning, at the time of diagnosis. aPLA values were also controlled after one year, and those values were correlated with the SLEDAI score values at that point, after one year. Also, the values of aPLA after three years, in the same patients, were correlated with the values of their SLEDAI scores after three years.

When talking about the correlation between the SLEDAI score and DI, these values were calculated for both, aPLA positive and aPLA negative groups of subjects, and then correlated at each time point (at the beginning, after one and after three years).

The correlation between SLEDAI-2K and DI at the beginning, at the time of diagnosis, was also made. As in the first year, there is no significant correlation between these parameters. Therefore, it is not shown in the Table V (but it is possible to display).

Thus, the correlation between aPLA and SLEDAI in Table IV, and between SLEDAI and DI in Table V, means that these are correlations of scores at each point.

According to your suggestion, I tried to give a different title to Table IV and V. I don't know if it's clearer now.

I apologize if I was too detailed. And once again, thank you very much for the suggestions.

Point 3 In line 22, you state, "it has been shown that aPLA positive patients have two and a half times higher risk of tissue damage than aPL negative ones." In Graphs 1 and 2, LN and CNS lupus tend to be more common in aPL- patients. Both data could be considered contradictory. That is, in the former, aPL+ can be interpreted as milder and in the latter, aPL+ as more severe. How do you explain this divergence in the data? One possibility that I have considered is that the aPL+ patients may have been treated more aggressively from the beginning because they had various organ lesions at the time of their first visit. This may have resulted in less tissue damage during follow-up. The current paper does not describe the details of the treatment, but I would appreciate it if you could take this into consideration and provide information that would be useful for the flow of the paper and for clinical practice.

Response 3

Patients with cSLE, aPLA positive were treated more aggressively. Higher doses of corticosteroids (pulse doses) were administered, antimalarials were given due to cytopenia, and in case of development of CNS manifestations, LN and cyclophosphamide, mycophenolate mofetil (with anticoagulant and antiplatelet therapy, according to indications).   In patients without aPLA, lower doses of corticosteroids were often sufficient to achieve remission, especially at the onset of the disease.

I don't know if the scope of this work would allow me to write in more detail about the administered doses of drugs. If you suggest that I can write about such details, I could include them in a few sentences.

Minor concerns

  • In line 35, what does "c-SLE" stand for, do you mean clinical SLE? If so, please add the abbreviation to the first occurrence for clarity. Also, since both cSLE and c-SLE appear, please unify the notation.

Response 1: Thank you. I corrected according to your suggestions. 

  • In line 81, "and15", please put a space between "and" and "15".

Response 2:  I inserted a space between “and” and “15”

  • In table 1, "SEL", do you mean SLE? Please correct accordingly.

Response 3: I’m sorry, it’s a mistake. SEL means SLE. I corrected according to your suggestions. 

  • In line 97, "Graph 1, 2" should be written in accordance with the submission rules, such as "Figure" instead of "Graph".

 Response 4: I corrected according to your suggestions, “Figure” instead “Graph” 

Figures 1 and Figure 2 can be attached in color, and can remain in black and white. I am sending the example in an additional document, where the corrected tables are also attached.

  • In Table I, II, 3, IV, V, numbers are not unified.

Response 5: I corrected numbers of tables: Table I, II, III, IV, V. Thank you. 

  • In Table IV, "SLEDAI scor" is considered to mean "score".

Response 6: I’m sorry, it’s a mistake. I corrected according to your suggestions. 

Tables and Figures

Dear Professor,

I apologize I am sending these tables and figures in a separate attachment, but I am afraid that some parts will be lost during the sending. I hope that didn't cause a problem.

Table II Presentation of clinical manifestations in the group of aPLA+ and aPLA - patients,

Clinical Manifestation

            aPLA positive

aPLA negative No(%) of Patients

Beg.             1.year               3.year

Beg            1.year                 3.year

Hematological disorders

Thrombocytopenia

8 (47,8)         4 (23,5)              4 (23,5)

7(29,4)       5 (21,7)            5 (21,7)

Leucopenia

5(29,4)          1 (5,9)                 2(11,7)

 9(39,1)      4(17,4)              2(8,69)

Autoimmune hemolytic anemia

7 (41,2)        3 (17,6)                 1 (5,9)             

12 (52)        9 (39)              4 (17,4)

Skin disorders

Livedo reticularis

3 (17,6)         5 (29,4)              9 (52,9)

3 (13)        6 ( 26)              11 (47,8)

Raynaud phenomen

5 (29,4)        8 (47)              11(64,7)

4(17,7)     7 (30,4)          12(52,2)

Neurologic disorders

Migraine headache

1(5,9)           4 (23,5)                2(11,8)

0                  2                      1 (4,35)

Epilepsy

0                 1 (5,9)                      1 (5,9)

0                  0                            0

Mood disorders

1 (5,9)         3 (17,6)                  1 (5,9)

0                  1 (4,35)           1 (4,35)

Table, suggestion

Table:  values of antiphospholipid antibodies of aPLA + patients

Patient No

 Sex (F/M)

LA

0        1y      3y

   aCLA IgM

0       1y       3y

aCLA IgG

0       1y       3y

β2GPI IgM

0 y     1y      3y

β2GPI IgG

0y     1y       3y

1   F

1,08    1     2,1

25,9  3,2    0,1

15,4  4,1   0,7

5,3   38      6,2

12    9,8     2,8

2   F

0,98    1     0,9

3,1    0,6    2,8

4,2   0,1   3,5

-         -        -

-        -        -

3   F

1,1     0,8  1,4

14,6   12    8,1

9,7   4,9    7,4

31     36      -

2,4   4,6     -

4   M

0,81   0,9  0,8

33,5   7,9   2,1

57,5  11,8  14

6,1    4,3     -

5,8   9,2     -

5   F

5,69  1,7  1,38

18    7,4   16,3

9,8   16,8   10

8,3    14,6  6,2

3,4  5,4   4,8

6   F

0,9     0,9   1,7

51    48    51,9

5,9   6,2   3,3

11,2  12   11,8

8,5   6,9   5,6

7   F

1,11   0,9   0,9

4,7  200   22,5

8,6   4,6   15,8

6,3  200  14,3

7,2   2,9   6,3

8   F

1,32  2,04  1,3

5,5   6,4    -

32,8  16     -

33,2  26   26,7

21,2  19   6,3

9   F

1,47  1,65  1,4

24   23,1  25,6

13,5  16  14,2

9,1   11,2  5,4

24,4  26  28,3

10  F

2,6    2,1    1,9

20,4  13   14,3

29,6  18  19,2

 9,8  6,8      8

16,3  12,3  4

11  M

1,1    1,5    1,6

23,5  18,5 6,5

7,5   6,2   4,8

-        -       -

-        -        -

12  F

1,8    2,34  2,8

120   62    24

110,9 51 21,4

38,4   9,2   5

19,2   4,3   6

13  M

0,9   1,8     2,1

20,7 13,7   14

50,7 25,6  28

19,6   12  14,2

4,7  5,4    7,8

14  M

0,96  1,1   0,8

10,7  12,6  11

17,5 16,8  18

3,5     4    3,6

6,1   4,2   5,6

15  M

2,3    3,1    0,6

-        -        -

-       -      -

-        -       -

-       -        -

16  F

0,8      1     0,9

17,1  8,4  12,6

12,6 10,8  18

6,4  18   19,2

4,2   5,1    4,4

17  M

1,1     0,8     1

6,2  12,2  18,4

16,2 22,2  22

18,2  19,2  18

21,8 16,2  25

0 - at diagnosis, 1y after one year of follow-up, 3y after three years of follow-up

aCLA,  β2GP (IgM, IgG) - PLU/ml, - not measured

Table IV Correlation of aPLA values with SLEDAI score during three-year follow-up

period of follow-up

Antibody

SLEDAI score

At the disease onset

LA

ρ=-0,145; p=0,386

aCLA IgM

ρ=0,054; p=0,747

aCLA IgG

ρ=0,170; p=0,307

Beta2 GPI IgM

ρ=0,352; p=0,035*

Beta2 GPI IgG

ρ=0,107; p=0,541

After the first year

LA

ρ=-0,263; p=0,110

aCLA IgM

ρ=0,096; p=0,595

aCLA IgG

ρ=0,276; p=0,121

Beta2 GPI IgM

ρ=0,341; p=0,070

Beta2 GPI IgG

ρ=-0,133; p=0,492

After the third year

LA

ρ=0,122; p=0,467

aCLA IgM

ρ=-0,160; p=0,415

aCLA IgG

ρ=-0,151; p=0,443

Beta2 GPI IgM

ρ=0,308; p=0,134

Beta2 GPI IgG

ρ=0,243; p=0,263

*statistically significant association; Spearman's correlation coefficient

Table V Association of SLEDAI-2K with DI at one- and three-year follow-up

Period of follow-up

Patient groups

p-value

Spearman’s correlation coefficient

The first year

Clinical SLE

0,508

-0,145

with APS

0,102

0,410

The third year

Clinical SLE

0,001

0,635

with APS

0,006

0,632

              t-test for 2 independent samples; Spearman’s correlation coefficient

Reviewer 2 Report

Major comments

The manuscript is neither clearly nor scientifically written. Very confusing paper. It is better to be stated that 40 patients consisted of 31 females and 9 males, and 17 aPL+(is this the same as with APS?) and 23 aPL- (is this the same as clinical SLE?). Definition of aPL+ is not clear. Does aPL+ mean all of lupus anticoagulant (LA)+ and anticardiolipin (aCL) + and anti-β2 GP1+ or either one of them at the time of diagnosis and 3 years later? In the Methods, it should be stated how LA, aCL-IgG, -IgM and anti-β2 GP1-IgG, -IgM were measured; qualitatively (pos or neg) or quantitatively (such as 1+, 2+ or 3+). Since the positive cases were only 17, new Table is better prepared in summarizing the distribution of 3 antibodies (pos or neg) for each of 17 cases, at the beginning (at diagnosis?), at first year and at third year. Consistency of group names is recommended, because clinical SLE is confusing, because all 40 patients were diagnosed as SLE.

Minor comments:

1.     Abbreviations: At first appearance, systemic lupus erythematosus needs to be abbreviated as SLE. Similarly, antiphospholipid antibodies at first appearance as aPLA (at one place aPLA+ and at others aPL+ ?). Same for others including aCLA (in Table 3, it was ACLA).

2.     Do children mean pediatric patients?

3.     Others

Line 34-35; c-SLE needs to be spelled out

Line 38; cSLE, not consistent with c-SLE

Line 62; positive values was equally and more than 1,2. (does not make sense)

Line 77; what does final group mean?

Table; comma (,) and period (.); 77.5% is not shown as 77,5%.

Table1; what is clinical SEL?

Table I title; Demographic characteristics of 40 SLE patients? Patients with aPL+ vs. Patients with aPL-?

Table II title: Comparison of clinical manifestations between aPL+ and aPL- group?

Table II; numbers are not on the same line. Difficult to interpret!

Graph 1 dotted line is used for aPL- while in Graph 2 solid line is used for aPL- (this is very confusing)

Line 108; all children were aPLA positive; all patients had aCLA (what does all mean?)

Line 111; all three antibodies had two children (11,76%) (This does not make sense)

Line 115; elevated class IgM was present in 11/16 patients (68,75%), and IgG in 10/16 patients (62,5%), (IgM or IgG class was not mentioned in the Methods)

In Table II, comparison was made between beginning (at diagnosis?) vs. 3 years and in Table 3, comparison is now made among beginning, first year and third year.

In Table 3, only β2GPI IgM is shown. On the other hand, in Table IV, β2GPI IgM and β2GPI IgG are shown (why?)

Line 157; 26% percent?

Line 167; lupus means SLE?

Line 170; how does cSLE differ from SLE with aPLA?

Line 171; istypes means isotypes?

Line 181; Campos et al. were not found; did not find?

Line 187; aPL negative children; aPL negative SLE patients?

Line 188; the presence of PL; PL or aPL?

Line 194; The presence of higher values of IgM; IgM-aCL? (what do higher values mean?)

Author Response

Dear Professor,
Thank you very much for all the suggestions. I tried to answer the questions and ambiguities. Also, since the correction of the English language was required, I requested the help of the official service of the journal platform (English Editing). I hope that the corrected text is more understandable. I am sending the following documents:
1) Response to your comments, major and minor
2) Corrected version of the text (in the attachment)
3) Proposal of modified tables and figures
4) English Editing Certificate

Thank you very much for all the information and help in reviewing my text.

Kind regards

Gordana Petrovic

Response to Reviewer 2 Comments

Major comments

Dear Professor, thank you very much for all the suggestions. I tried to correct the article, thanks to your comments, to make it understandable for the reader.  For more precise answers, I have divided your major comment into several smaller parts. I hope that's okay? The answers are below.

The manuscript is neither clearly nor scientifically written. Very confusing paper. It is better to be stated that 40 patients consisted of 31 females and 9 males, and 17 aPL+(is this the same as with APS?) and 23 aPL- (is this the same as clinical SLE?). Definition of aPL+ is not clear. Does aPL+ mean all of lupus anticoagulant (LA)+ and anticardiolipin (aCL) + and anti-β2 GP1+ or either one of them at the time of diagnosis and 3 years later? In the Methods, it should be stated how LA, aCL-IgG, -IgM and anti-β2 GP1-IgG, -IgM were measured; qualitatively (pos or neg) or quantitatively (such as 1+, 2+ or 3+). Since the positive cases were only 17, new Table is better prepared in summarizing the distribution of 3 antibodies (pos or neg) for each of 17 cases, at the beginning (at diagnosis?), at first year and at third year. Consistency of group names is recommended, because clinical SLE is confusing, because all 40 patients were diagnosed as SLE.

Responses

The manuscript is neither clearly nor scientifically written. Very confusing paper. It is better to be stated that 40 patients consisted of 31 females and 9 males, and 17 aPL+(is this the same as with APS?) and 23 aPL- (is this the same as clinical SLE?).

Response: Thank you. I corrected according to your suggestions.

Definition of aPL+ is not clear. Does aPL+ mean all of lupus anticoagulant (LA)+ and anticardiolipin (aCL) + and anti-β2 GP1+ or either one of them at the time of diagnosis and 3 years later?

Response: aPLA + is not the same as APS. aPLA - is a group of clinical SLE, because of easier understanding of the text, according to your suggestions, I would suggest that the groups be called aPLA positive, or aPLA + (17 patients) and aPLA negative, or aPLA - (23 patients). An aPLA positive patient means the presence of at least one aPLA at the time of diagnosis (at the beginning), after one and three years of follow-up.

 In the Methods, it should be stated how LA, aCL-IgG, -IgM and anti-β2 GP1-IgG, -IgM were measured; qualitatively (pos or neg) or quantitatively (such as 1+, 2+ or 3+).

Response: In the Methods, according to your suggestions, I listed the threshold values from all aPLA. I am sending you the corrected section below. If necessary, I can add more details about these antibodies.

“The lupus anticoagulans (LA) in the samples of each enrolled patient was tested and positive values was  ≥1,2. For the detection of LA antibodies, the dilute Russel viper venom test (DRVVT) was applied. Anticardiolipin (aCLA) and β2GPI antibodies were tested by Enzyme Linked Immuno Sorbent Assay (ELISA) technique (Euroimmun, Mediziniche Labordiagnostika AG) and the results were expressed in international units (PLU/ml). They are defined as positive if aCLA (IgM/IgG) is >12 PLU/ml. Values less than 20 PLU/ml are weakly positive, from 20 to 40 PLU/ml moderately positive, and above 40 PLU/ml strongly positive. β2GPI positive values were those above 10 U/ml”

Since the positive cases were only 17, new Table is better prepared in summarizing the distribution of 3 antibodies (pos or neg) for each of 17 cases, at the beginning (at diagnosis?), at first year and at third year.

Table, suggestion (according to the suggestion of the 2nd reviewer)

Table:  values of antiphospholipid antibodies of aPLA + patients

Patient No

 Sex (F/M)

LA

0        1y      3y

   aCLA IgM

0       1y       3y

aCLA IgG

0       1y       3y

β2GPI IgM

0 y     1y      3y

β2GPI IgG

0y     1y       3y

1   F

1,08    1     2,1

25,9  3,2    0,1

15,4  4,1   0,7

5,3   38      6,2

12    9,8     2,8

2   F

0,98    1     0,9

3,1    0,6    2,8

4,2   0,1   3,5

-         -        -

-        -        -

3   F

1,1     0,8  1,4

14,6   12    8,1

9,7   4,9    7,4

31     36      -

2,4   4,6     -

4   M

0,81   0,9  0,8

33,5   7,9   2,1

57,5  11,8  14

6,1    4,3     -

5,8   9,2     -

5   F

5,69  1,7  1,38

18    7,4   16,3

9,8   16,8   10

8,3    14,6  6,2

3,4  5,4   4,8

6   F

0,9     0,9   1,7

51    48    51,9

5,9   6,2   3,3

11,2  12   11,8

8,5   6,9   5,6

7   F

1,11   0,9   0,9

4,7  200   22,5

8,6   4,6   15,8

6,3  200  14,3

7,2   2,9   6,3

8   F

1,32  2,04  1,3

5,5   6,4    -

32,8  16     -

33,2  26   26,7

21,2  19   6,3

9   F

1,47  1,65  1,4

24   23,1  25,6

13,5  16  14,2

9,1   11,2  5,4

24,4  26  28,3

10  F

2,6    2,1    1,9

20,4  13   14,3

29,6  18  19,2

 9,8  6,8      8

16,3  12,3  4

11  M

1,1    1,5    1,6

23,5  18,5 6,5

7,5   6,2   4,8

-        -       -

-        -        -

12  F

1,8    2,34  2,8

120   62    24

110,9 51 21,4

38,4   9,2   5

19,2   4,3   6

13  M

0,9   1,8     2,1

20,7 13,7   14

50,7 25,6  28

19,6   12  14,2

4,7  5,4    7,8

14  M

0,96  1,1   0,8

10,7  12,6  11

17,5 16,8  18

3,5     4    3,6

6,1   4,2   5,6

15  M

2,3    3,1    0,6

-        -        -

-       -      -

-        -       -

-       -        -

16  F

0,8      1     0,9

17,1  8,4  12,6

12,6 10,8  18

6,4  18   19,2

4,2   5,1    4,4

17  M

1,1     0,8     1

6,2  12,2  18,4

16,2 22,2  22

18,2  19,2  18

21,8 16,2  25

0 - at diagnosis, 1y after one year of follow-up, 3y after three years of follow-up

aCLA,  β2GP (IgM, IgG) - PLU/ml, - not measured

Consistency of group names is recommended, because clinical SLE is confusing, because all 40 patients were diagnosed as SLE.

Thank you for your suggestions. Indeed, there are mistakes in writing abbreviations in my text, which significantly complicates the understanding of the text. I tried to adequately introduce abbreviations, and to use them correctly throughout the text. I have attached the corrected version of the work.

Abbreviations: SLE - Systemic Lupus Erythematosus, cSLE - childhood onset Systemic Lupus Erythematosus, aPLA - antiphospholipid antibodies, aCLA - anticardiolipin antibodies, β2GPI - β2glycoprotein I antibodies, LA - lupus anticoagulans

The aim of the study is to evaluate the effect of the presence of antiphospholipid antibodies on clinical and laboratory manifestations, activity and outcome of the disease in patients with childhood-onset systemic lupus erythematosus (cSLE)

Minor comments

  1. Abbreviations: At first appearance, systemic lupus erythematosus needs to be abbreviated as SLE. Similarly, antiphospholipid antibodies at first appearance as aPLA (at one place aPLA+ and at others aPL+ ?). Same for others including aCLA (in Table 3, it was ACLA).

Response  1. Dear Professor, thank you for your suggestions. Indeed, there are mistakes in writing abbreviations in my text, which significantly complicates the understanding of the text. I tried to adequately introduce abbreviations, and to use them correctly throughout the text. I have attached the corrected version of the article.

Abbreviations:

SLE - Systemic Lupus Erythematosus

cSLE - childhood onset Systemic Lupus Erythematosus

aPLA - antiphospholipid antibodies

aCLA - anticardiolipin antibodies

β2GPI - β2glycoprotein I antibodies

LA - lupus anticoagulans

  1. Do children mean pediatric patients?

Response 2. Yes, I wrote "children" several times in the text, referring to pediatric patients with SLE. I corrected, according to your suggestion, so that it could be better understood.

3.Others

Response 3. I did not understand this suggestion

Line 34-35; c-SLE needs to be spelled out

Line 38; cSLE, not consistent with c-SLE

Response Lines 34, 35, 38: I changed the abbreviations and entered the correct ones accordingly.

Line 62; positive values was equally and more than 1,2. (does not make sense)

Response Line 62: I put an adequate mark (  ≥ ).

Line 77; what does final group mean?

Response Line 77. "final group" includes aPLA positive and aPLA negative patients with systemic lupus erythematosus. I added that in the sentence. Thank you for your comment.

Table; comma (,) and period (.); 77.5% is not shown as 77,5%.

Response Table. I have corrected this error. Thanks for the suggestion.

Table1; what is clinical SEL?

Response Table 1. Clinical SLE" is an aPLA negative group of patients with cSLE. I corrected it in the table.

Table I title; Demographic characteristics of 40 SLE patients? Patients with aPL+ vs. Patients with aPL-?

Response Table I Title. Yes, those are the demographic characteristics of aPLA+ and aPLA - patients.

Do you think it is correct that the title of Table1 should be: "Demographic characteristics of aPLA+ and aPLA- patients?

Table II title: Comparison of clinical manifestations between aPL+ and aPL- group?

Response Table II title. Table II shows the clinical manifestations of the aPLA+ and aPLA - groups. It might be correct that the title should read: "presentation of clinical manifestations in the group of aPLA+ and aPLA - patients"

Table II; numbers are not on the same line. Difficult to interpret!

Response Table II.Table II lost certain parts in the transmission of the manuscript. I will send you a slightly different version, I don't know if  this one is more adequate?

Table II Presentation of clinical manifestations in the group of aPLA+ and aPLA - patients"

Clinical Manifestation

            aPLA positive

aPLA negative No(%) of Patients

Beg.             1.year               3.year

Beg            1.year                 3.year

Hematological disorders

Thrombocytopenia

8 (47,8)         4 (23,5)              4 (23,5)

7(29,4)       5 (21,7)            5 (21,7)

Leucopenia

5(29,4)          1 (5,9)                 2(11,7)

 9(39,1)      4(17,4)              2(8,69)

Autoimmune hemolytic anemia

7 (41,2)        3 (17,6)                 1 (5,9)             

12 (52)        9 (39)              4 (17,4)

Skin disorders

Livedo reticularis

3 (17,6)         5 (29,4)              9 (52,9)

3 (13)        6 ( 26)              11 (47,8)

Raynaud phenomen

5 (29,4)        8 (47)              11(64,7)

4(17,7)     7 (30,4)          12(52,2)

Neurologic disorders

Migraine headache

1(5,9)           4 (23,5)                2(11,8)

0                  2                      1 (4,35)

Epilepsy

0                 1 (5,9)                      1 (5,9)

0                  0                            0

Mood disorders

1 (5,9)         3 (17,6)                  1 (5,9)

0                  1 (4,35)           1 (4,35)

Graph 1 dotted line is used for aPL- while in Graph 2 solid line is used for aPL- (this is very confusing)

Response Graph1.  Thank you for this observation. In the Figures, the aPLA + group is shown with a solid line, and the aPLA - group with a dotted line. I have these Figures  in color.  If it is clearer, I am sending these figures in color. aPLA + group is shown in blue and aPLA - in red color. Perhaps they are easier to understand. I am sending them below.

Graph 2 The probability of the occurrence of the CNS complications and the clinical form of the disease.

Graph 1. The probability of the occurrence of the lupus nephritis and the clinical form of the disease.

Line 108; all children were aPLA positive; all patients had aCLA (what does all mean?)

Response Line 108. This means that all patients who had CNS manifestations had at least one of the antiphospholipid antibodies, they were from the aPLA + group. All patients with SLE and CNS manifestations had anticardiolipin antibodies, patient with chorea had LA, children with vasculitis had both LA and β2GP1.

Line 111; all three antibodies had two children (11,76%) (This does not make sense)

Response Line 111. Maybe it can be said like this: "Two children (11,76%) had all antiphospholipid antibodies"

Line 115; elevated class IgM was present in 11/16 patients (68,75%), and IgG in 10/16 patients (62,5%), (IgM or IgG class was not mentioned in the Methods)

Response Line 115. Thank you. After your comment, I corrected and explained it in the Methods

In Table II, comparison was made between beginning (at diagnosis?) vs. 3 years and in Table 3, comparison is now made among beginning, first year and third year.

Response Table II. I have attached the corrected table II, with values entered even after the 1st year of follow-up, in the previous answers. I appreciate your suggestions regarding this modified table. Thank you very much.

In Table 3, only β2GPI IgM is shown. On the other hand, in Table IV, β2GPI IgM and β2GPI IgG are shown (why?)

Response In Table 3. “Table 3” is an error, I corrected it in “Table III”.  In Table IV, only β2GPI IgM is shown because significance was determined only in the correlation of the concentration of this antibody with the SLEDAI score. However, I have a table showing the correlation of all antibodies with the SLEDAI score and I am attaching it below.

Table IV Correlation of aPLA with SLEDAI score during three-year follow-up

period of follow-up

Antibody

SLEDAI score

At the disease onset

LA

ρ=-0,145; p=0,386

aCLA IgM

ρ=0,054; p=0,747

aCLA IgG

ρ=0,170; p=0,307

Beta2 GPI IgM

ρ=0,352; p=0,035*

Beta2 GPI IgG

ρ=0,107; p=0,541

After the first year

LA

ρ=-0,263; p=0,110

aCLA IgM

ρ=0,096; p=0,595

aCLA IgG

ρ=0,276; p=0,121

Beta2 GPI IgM

ρ=0,341; p=0,070

Beta2 GPI IgG

ρ=-0,133; p=0,492

After the third year

LA

ρ=0,122; p=0,467

aCLA IgM

ρ=-0,160; p=0,415

aCLA IgG

ρ=-0,151; p=0,443

Beta2 GPI IgM

ρ=0,308; p=0,134

Beta2 GPI IgG

ρ=0,243; p=0,263

*statistically significant association; Spearman's correlation coefficient

Line 157; 26% percent?

Response Line 157. Yes, the word “percent” is not needed. I corrected.

Line 167; lupus means SLE?

Response Line 167. Yes, that's a mistake. Lupus means SLE. I corrected.

Line 170; how does cSLE differ from SLE with aPLA?

Response 170. cSLE is Systemic Lupus Erythematosus with childhood onset. In these children, aPLA can also occur. If in a repeated sample, after 12 weeks from the first detection of aPLA (at least one of LA, aCLA or  β2GPI), we again determine the presence of one of the aPLA, the patient is considered to be aPLA positive. A child with SLE (who has cSLE) who does not have aPLA (at least one of the antibodies) is considered an aPLA negative cSLE patient.

Line 171; istypes means isotypes? 

Response Line 171. That's a mistake. I corrected.

Line 181; Campos et al. were not found; did not find?

Response Line 181. I'm sorry, the reference number is wrong. It is reference number 12 (not number 16) I also corrected it in my article.

Line 187; aPL negative children; aPL negative SLE patients?

Response Line 187. Yes, I meant SLE patients. I corrected, thank you very much.

Line 188; the presence of PL; PL or aPL?

Response Line 188. That's a mistake. I corrected.

Line 194; The presence of higher values of IgM; IgM-aCL? (what do higher values mean?)

Response Line 194. In the Methods, according to your suggestions, I listed the threshold values from all aPLA. I am sending you the corrected section below. If necessary, I can add more details about these antibodies.

“The lupus anticoagulans (LA) in the samples of each enrolled patient was tested and positive values was  ≥1,2. For the detection of LA antibodies, the dilute Russel viper venom test (DRVVT) was applied. Anticardiolipin (aCLA) and β2GPI antibodies were tested by Enzyme Linked Immuno Sorbent Assay (ELISA) technique (Euroimmun, Mediziniche Labordiagnostika AG) and the results were expressed in international units (PLU/ml). They are defined as positive if aCLA (IgM/IgG) is >12 PLU/ml. Values less than 20 PLU/ml are weakly positive, from 20 to 40 PLU/ml moderately positive, and above 40 PLU/ml strongly positive. β2GPI positive values were those above 10 U/ml”

Dear Professor,

I apologize I am sending these tables and figures in a separate attachment, but I am afraid that some parts will be lost during the sending. I hope that didn't cause a problem.

Table II Presentation of clinical manifestations in the group of aPLA+ and aPLA - patients,

Clinical Manifestation

            aPLA positive

aPLA negative No(%) of Patients

Beg.             1.year               3.year

Beg            1.year                 3.year

Hematological disorders

Thrombocytopenia

8 (47,8)         4 (23,5)              4 (23,5)

7(29,4)       5 (21,7)            5 (21,7)

Leucopenia

5(29,4)          1 (5,9)                 2(11,7)

 9(39,1)      4(17,4)              2(8,69)

Autoimmune hemolytic anemia

7 (41,2)        3 (17,6)                 1 (5,9)             

12 (52)        9 (39)              4 (17,4)

Skin disorders

Livedo reticularis

3 (17,6)         5 (29,4)              9 (52,9)

3 (13)        6 ( 26)              11 (47,8)

Raynaud phenomen

5 (29,4)        8 (47)              11(64,7)

4(17,7)     7 (30,4)          12(52,2)

Neurologic disorders

Migraine headache

1(5,9)           4 (23,5)                2(11,8)

0                  2                      1 (4,35)

Epilepsy

0                 1 (5,9)                      1 (5,9)

0                  0                            0

Mood disorders

1 (5,9)         3 (17,6)                  1 (5,9)

0                  1 (4,35)           1 (4,35)

Table, suggestion

Table:  values of antiphospholipid antibodies of aPLA + patients

Patient No

 Sex (F/M)

LA

0        1y      3y

   aCLA IgM

0       1y       3y

aCLA IgG

0       1y       3y

β2GPI IgM

0 y     1y      3y

β2GPI IgG

0y     1y       3y

1   F

1,08    1     2,1

25,9  3,2    0,1

15,4  4,1   0,7

5,3   38      6,2

12    9,8     2,8

2   F

0,98    1     0,9

3,1    0,6    2,8

4,2   0,1   3,5

-         -        -

-        -        -

3   F

1,1     0,8  1,4

14,6   12    8,1

9,7   4,9    7,4

31     36      -

2,4   4,6     -

4   M

0,81   0,9  0,8

33,5   7,9   2,1

57,5  11,8  14

6,1    4,3     -

5,8   9,2     -

5   F

5,69  1,7  1,38

18    7,4   16,3

9,8   16,8   10

8,3    14,6  6,2

3,4  5,4   4,8

6   F

0,9     0,9   1,7

51    48    51,9

5,9   6,2   3,3

11,2  12   11,8

8,5   6,9   5,6

7   F

1,11   0,9   0,9

4,7  200   22,5

8,6   4,6   15,8

6,3  200  14,3

7,2   2,9   6,3

8   F

1,32  2,04  1,3

5,5   6,4    -

32,8  16     -

33,2  26   26,7

21,2  19   6,3

9   F

1,47  1,65  1,4

24   23,1  25,6

13,5  16  14,2

9,1   11,2  5,4

24,4  26  28,3

10  F

2,6    2,1    1,9

20,4  13   14,3

29,6  18  19,2

 9,8  6,8      8

16,3  12,3  4

11  M

1,1    1,5    1,6

23,5  18,5 6,5

7,5   6,2   4,8

-        -       -

-        -        -

12  F

1,8    2,34  2,8

120   62    24

110,9 51 21,4

38,4   9,2   5

19,2   4,3   6

13  M

0,9   1,8     2,1

20,7 13,7   14

50,7 25,6  28

19,6   12  14,2

4,7  5,4    7,8

14  M

0,96  1,1   0,8

10,7  12,6  11

17,5 16,8  18

3,5     4    3,6

6,1   4,2   5,6

15  M

2,3    3,1    0,6

-        -        -

-       -      -

-        -       -

-       -        -

16  F

0,8      1     0,9

17,1  8,4  12,6

12,6 10,8  18

6,4  18   19,2

4,2   5,1    4,4

17  M

1,1     0,8     1

6,2  12,2  18,4

16,2 22,2  22

18,2  19,2  18

21,8 16,2  25

0 - at diagnosis, 1y after one year of follow-up, 3y after three years of follow-up

aCLA,  β2GP (IgM, IgG) - PLU/ml, - not measured

Table IV Correlation of aPLA values with SLEDAI score during three-year follow-up

period of follow-up

Antibody

SLEDAI score

At the disease onset

LA

ρ=-0,145; p=0,386

aCLA IgM

ρ=0,054; p=0,747

aCLA IgG

ρ=0,170; p=0,307

Beta2 GPI IgM

ρ=0,352; p=0,035*

Beta2 GPI IgG

ρ=0,107; p=0,541

After the first year

LA

ρ=-0,263; p=0,110

aCLA IgM

ρ=0,096; p=0,595

aCLA IgG

ρ=0,276; p=0,121

Beta2 GPI IgM

ρ=0,341; p=0,070

Beta2 GPI IgG

ρ=-0,133; p=0,492

After the third year

LA

ρ=0,122; p=0,467

aCLA IgM

ρ=-0,160; p=0,415

aCLA IgG

ρ=-0,151; p=0,443

Beta2 GPI IgM

ρ=0,308; p=0,134

Beta2 GPI IgG

ρ=0,243; p=0,263

*statistically significant association; Spearman's correlation coefficient

Table V Association of SLEDAI-2K with DI at one- and three-year follow-up

Period of follow-up

Patient groups

p-value

Spearman’s correlation coefficient

The first year

Clinical SLE

0,508

-0,145

with APS

0,102

0,410

The third year

Clinical SLE

0,001

0,635

with APS

0,006

0,632

              t-test for 2 independent samples; Spearman’s correlation coefficient

Reviewer 3 Report

The manuscript reports the onset of clinical manifestations in children with systemic lupus erythematosus and anti-phospholipid antibodes.

The topic may be of potential interest for clinicians, however the English language is very poor and requires a complete revision by a native English to improve the understanding of the results. Moreover, tables contain many typesetting errors and need a revision.

Author Response

Dear Professor,
Thank you very much for all the suggestions. I tried to answer the questions and ambiguities. Also, since the correction of the English language was required, I requested the help of the official service of the journal platform (English Editing). I hope that the corrected text is more understandable. I am sending the following documents in the attachment:
1) response to your comments, proposal of modified tables and figures
2) corrected version of the text
3) English Editing Certificate

Thank you very much for all the information and help in reviewing my text.

Kind regards

Gordana Petrovic

Round 2

Reviewer 1 Report

The authors have responded and corrected as appropriate. There are no additional matters to be addressed.

Author Response

Dear Professor,

Thanks for the suggestions and help.

Sincerely

Gordana Petrovic

Reviewer 2 Report

Major comments

Further refinement of the manuscript (including grammatical corrections) is required. Also, this is not a complete revised manuscript, which seems still in processing. The authors did not yet decide how to manage additional Table (Table, according to the reviewer’s suggestion should not be stated within the text). Now, Tables are 5 or 6? The authors must decide if additional Table is appropriate as new Table II or put it into Supplemental Table). Results should be more clearly and concisely stated. Once again, this reviewer would like to emphasize the use of period (.) than comma (,) for % in all text and in Tables, 77,4 % (wrong) and 77.4% (right). In the text, why mix of children and patients were used? Mix of cSLE and pediatric SLE are also employed. Mix of patients with cSLE and children with SLE are employed.

Minor comments:

1.     Reference numbers should be placed at the end of sentence on top right?

2.     Key words: children rheumatology should be pediatric rheumatology.

3.     In Tables, (0, 1y, 3y), (Beg. 1.year, 3.year), and (the disease onset, the first year, the third year) needs consistency; why no 0 or disease onset in Table V?

4.     Table II, numbers are enough, (%) is not necessary, because numbers are so small.

5.     In Figure 1 (× is for aPL-), while in Figure 2 (× is for aPL+). This is confusing.

6.     Line 123, the highest in the first four years of follow-up, (The patients were followed up to 3 years? Four years of follow-up sound peculiar to readers?)

7.     Line 133; aCLA was not performed in 1 patients ?

8.     Line 140, antiphospholipid antibody values; antiphospholipid antibody has already been abbreviated as aPL?

9.     Line 177. seven and six patient respectively, patients?

Author Response

Dear Professor,
Thank you very much for all the suggestions. I tried to answer the questions and ambiguities. Also, since the correction of the English language was required, I requested the help of the official service of the journal platform (English Editing). I hope that the corrected text is more understandable. I am sending the following documents in the meil and in the attachment:
1) response to your comments, major and minor
2) corrected version of the text
3) Supplemental Table, Table S1

Thank you very much for all the information and help in reviewing my text.

Kind regards

Gordana Petrovic

Response to Reviewer 2 Comments, Round 2

Major comments

Dear Professor, thank you very much for all the suggestions. I tried to correct the article, thanks to your comments, to make it understandable for the reader.  For more precise answers, I have divided your major comment into several smaller parts. I hope that's okay? The answers are below.

Further refinement of the manuscript (including grammatical corrections) is required. Also, this is not a complete revised manuscript, which seems still in processing. The authors did not yet decide how to manage additional Table (Table, according to the reviewer’s suggestion should not be stated within the text). Now, Tables are 5 or 6? The authors must decide if additional Table is appropriate as new Table II or put it into Supplemental Table). Results should be more clearly and concisely stated. Once again, this reviewer would like to emphasize the use of period (.) than comma (,) for % in all text and in Tables, 77,4 % (wrong) and 77.4% (right). In the text, why mix of children and patients were used? Mix of cSLE and pediatric SLE are also employed.

Responses

Further refinement of the manuscript (including grammatical corrections) is required. Also, this is not a complete revised manuscript, which seems still in processing. The authors did not yet decide how to manage additional Table (Table, according to the reviewer’s suggestion should not be stated within the text). Now, Tables are 5 or 6? The authors must decide if additional Table is appropriate as new Table II or put it into Supplemental Table)

Response: Thank you. I corrected according to your suggestions. The paper was also reviewed by the English Editing service, the journal's platform. An additional table related to aPLA values is defined as Supplemental Table 1 (Table S1). In the text, in line 142, I added a sentence “The aPLA values of our patients are given in Table S1”.

 Other tables have kept the same numbers and titles.

Once again, this reviewer would like to emphasize the use of period (.) than comma (,) for % in all text and in Tables, 77,4 % (wrong) and 77.4% (right).

Response: Thank you. I corrected according to your suggestions, in all text and in Tables.

In the text, why mix of children and patients were used? Mix of cSLE and pediatric SLE are also employed. Mix of patients with cSLE and children with SLE are employed.

Response: Thank you. I corrected according to your suggestions, in all text.

Minor comments:

  1. Reference numbers should be placed at the end of sentence on top right?

Response 1: In the Author's Guidelines is recommanded: " In the text, reference numbers should be placed in square brackets [ ], and placed before the punctuation; for example [1], [1–3] or [1,3]. I listed the reference numbers according to these instructions.

  1. Key words: children rheumatology should be pediatric rheumatology.

Response 2: Thank you. I corrected according to your suggestions.

  1.  In Tables, (0, 1y, 3y), (Beg. 1.year, 3.year), and (the disease onset, the first year, the third year) needs consistency; why no 0 or disease onset in Table V?

Response 3:  I corrected according to your suggestions. Table V does not show the correlation between SLEDAI and DI at the beginning, at the time of diagnosis, because no significant correlation was established, nor after the first year. I also added data for the onset of the disease. Thank you for the suggestion.

  1. Table II, numbers are enough, (%) is not necessary, because numbers are so small.

Response 4: Thank you. I corrected according to your suggestions.

  1. In Figure 1 (× is for aPL-), while in Figure 2 (× is for aPL+). This is confusing.

Response 5:  It's a cross and star for censored patients... or patients without complications. These figures can also be shown in color, if you think it is clear way. They are show in the text.

6. Line 123, the highest in the first four years of follow-up, (The patients were followed up to 3 years? Four years of follow-up sound peculiar to readers?)

Response 6: In the “ Materials and Methods” section, we stated that the measurements of parameters (clinical and laboratory) were performed at the time of diagnosis of the disease, after one and three years of follow-up. However, since we were able to use the Kaplan-Mayer method to estimate the probability of the occurrence of complications in a longer follow-up patients ( in the Materials and Methods write “The course of the disease was followed up over the period of ten years through clinical and laboratory parameters”) we thought that readers would find these data meaningful.

  1. Line 133; aCLA was not performed in 1 patients ?

Response 7: Thank you. I corrected according to your suggestions.

  1. Line 140, antiphospholipid antibody values; antiphospholipid antibody has already been abbreviated as aPL?

Response 8: Thank you. I corrected according to your suggestions.

  1. Line 177. seven and six patient respectively, patients?

Response 9: Thank you. I corrected according to your suggestions.

Table S1 (ex “Table, according to the reviewer’s suggestion”)

This table will be supplemental, Table S1

Table S1: values of antiphospholipid antibodies of aPLA + patients

Patient No

 Sex (F/M)

LA

beg   1.y    3.y

   aCLA IgM

beg   1.y    3y

aCLA IgG

beg   1.y    3.y

β2GPI IgM

beg   1.y    3.y

β2GPI IgG

beg  1.y    3.y

1   F

1.08   1     2.1

25.9  3.2    0.1

15.4  4.1   0.7

5.3   38     6.2

12   9.8     2.8

2   F

0.98   1     0.9

3.1    0.6    2.8

4.2   0.1   3.5

-     -        -

-     -        -

3   F

1.1    0.8   1.4

14.6   12    8.1

9.7   4.9    7.4

31     36      -

2.4   4.6     -

4   M

0.81   0.9   0.8

33.5   7.9   2.1

57.5  11.8  14

6.1    4.3     -

5.8   9.2     -

5   F

5.69  1.7   1.38

18    7.4   16.3

9.8   16.8   10

8.3    14.6  6.2

3.4   5.4   4.8

6   F

0.9    0.9   1.7

51    48    51.9

5.9   6.2   3.3

11.2   12   11.8

8.5   6.9   5.6

7   F

1.11   0.9   0.9

4.7  200    22.5

8.6   4.6   15.8

6.3    200  14.3

7.2   2.9   6.3

8   F

1.32   2.04  1.3

5.5   6.4    -

32.8  16     -

33.2   26   26.7

21.2   19    6.3

9   F

1.47   1.65  1.4

24   23.1  25.6

13.5  16    14.2

9.1    11.2  5.4

24.4   26   28.3

10  F

2.6    2.1    1.9

20.4  13   14.3

29.6  18    19.2

 9.8   6.8     8

16.3   12.3   4

11  M

1.1    1.5    1.6

23.5  18.5  6.5

7.5   6.2    4.8

-      -       -

-     -        -

12  F

1.8    2.34  2.8

120   62    24

110.9  51   21.4

38.4    9.2   5

19.2   4.3   6

13  M

0.9    1.8    2.1

20.7  13.7   14

50.7  25.6   28

19.6   12   14.2

4.7    5.4    7.8

14  M

0.96   1.1   0.8

10.7  12.6   11

17.5  16.8   18

3.5     4    3.6

6.1    4.2   5.6

15  M

2.3    3.1    0.6

-     -        -

-       -      -

-      -       -

-     -        -

16  F

0.8    1     0.9

17.1   8.4  12.6

12.6   10.8  18

6.4    18   19.2

4.2    5.1    4.4

17  M

1.1    0.8     1

6.2   12.2  18.4

16.2   22.2  22

18.2    19.2  18

21.8   16.2   25

beg - at the beginning,  1.y after one year of follow-up, 3.y after three years of follow-up

aCLA, β2GP (IgM, IgG) - PLU/ml, - not measured

Reviewer 3 Report

The revised manuscript may be accepted in present form

Author Response

(The authors gave the same response as above.)
